Effectiveness of seed dispersal by foxes in areas with different human disturbances in southern Chile

Triay-Limonta Onaylis onaylis.triay@alumnos.ulagos.cl 1 2
Paleo-López Rocío 1 2
Stuardo Camila J. 1
Ugarte Carolina S. 1 2
Valdivia Carlos E. 3
Napolitano Constanza constanza.napolitano@ulagos.cl 1 4 5
1 Laboratorio de Genética de la Conservación, Departamento de Ciencias Biológicas y Biodiversidad, Universidad de Los Lagos , Osorno , Los Lagos Región , Chile
2 Programa de Doctorado en Ciencias, mención Conservación y Manejo de Recursos Naturales, Universidad de Los Lagos , Osorno , Los Lagos Región , Chile
3 Laboratorio de Vida Silvestre, Departamento de Ciencias Biológicas y Biodiversidad, Universidad de Los Lagos , Osorno , Los Lagos Región , Chile
4 Institute of Ecology and Biodiversity , Concepción , Región Gran Concepción , Chile
5 Cape Horn International Center , Puerto Williams , Región Magallanes y Antártida , Chile
Manjarrez Javier
Electronic publication date: 2025 Oct 21
Publication date: 2025
Volume: 13
Electronic Location ID: e20150
Received 2025 Feb 21; Accepted 2025 Sep 5
Copyright: ©2025 Triay-Limonta et al.
Copyright year: 2025
Copyright holder: Triay-Limonta et al.
License: This is an open access article distributed under the terms of the Creative Commons Attribution License, which permits unrestricted use, distribution, reproduction and adaptation in any medium and for any purpose provided that it is properly attributed. For attribution, the original author(s), title, publication source (PeerJ) and either DOI or URL of the article must be cited.
License URL: https://creativecommons.org/licenses/by/4.0/

Keywords: Seed dispersal effectiveness (SDE), Lycalopex spp., Frugivory, Human disturbance, Native plants, Exotic species, Southern Chile, Temperate forest

Funding: The ANID Fondecyt Regular 1220758 and 1251063, and ANID PIA/BASAL FB210006, ANID/BASAL FB210018 Onaylis Triay-Limonta, Rocío Paleo-López and Carolina S. Ugarte received scholarships for doctoral studies (Onaylis Triay-Limonta, doctoral scholarship ANID no 21210070; Rocío Paleo-López, doctoral scholarship ANID no 21210979; Carolina S. Ugarte doctoral scholarship ANID no 21220797) from the Programa de Doctorado en Ciencias, mención Conservación y Manejo de Recursos Naturales, Universidad de Los Lagos The Chilean National Agency for Research and Development [ANID, Chile] This work was supported by the ANID Fondecyt Regular 1220758 and 1251063, and ANID PIA/BASAL FB210006, ANID/BASAL FB210018. Onaylis Triay-Limonta, Rocío Paleo-López and Carolina S. Ugarte received scholarships for doctoral studies (Onaylis Triay-Limonta, doctoral scholarship ANID no 21210070; Rocío Paleo-López, doctoral scholarship ANID no 21210979; Carolina S. Ugarte doctoral scholarship ANID no 21220797) from the Programa de Doctorado en Ciencias, mención Conservación y Manejo de Recursos Naturales, Universidad de Los Lagos, and the Chilean National Agency for Research and Development [ANID, Chile]. The funders had no role in study design, data collection and analysis, decision to publish, or preparation of the manuscript.

==============================
Frugivorous mammals play a key role in forest regeneration by dispersing seeds yet their effectiveness can vary with landscape disturbance and the native or exotic status of the plant species. In the temperate forests of southern Chile, we evaluated the seed dispersal efficiency (SDE) of foxes (Lycalopex spp.) for native and exotic fleshy-fruited plants across 10 sites spanning a gradient from less to more disturbed landscapes. SDE was assessed though two components: quantity (frequency of feces containing seeds and number of seeds per plant species) and quality (proportion of viable seeds and their germination success under natural conditions). Between November 2022 and July 2023, we collected 199 fox fecal samples, of which 131 contained seeds. In total, we recovered of 23,012 seeds from three native species (Aristotelia chilensis, Gaultheria mucronata, Ugni molinae) and three exotic species (Prunus spp., Malus sylvestris, Rubus ulmifolius). Native seeds were more frequently found in feces from less disturbed landscapes, whereas exotic species exhibited more variable patterns. Seed viability was generally high and unaffected by landscape type, although germination rates were higher in less disturbed areas. Notably, G. mucronata failed to germinate in more disturbed landscapes despite high viability. Conversely, U. molinae showed significantly greater SDE in more disturbed areas due to higher seed quantity and germination. These findings demonstrate that foxes are effective seed dispersers of both native and exotic species, with their effectiveness shaped by landscape type and plant species. Our results highlight the dual ecological role of generalist frugivores in supporting native plant regeneration while potentially facilitating the spread of exotic species. We recommend that future conservation and restoration efforts consider these dynamics when managing fragmented temperate forests.

Introduction

Understanding mutualistic interactions between plants and animals is crucial for biodiversity conservation, especially in the current context of increasing anthropogenic disturbances (Fricke et al., 2013; Levi & Peres, 2013). Human activities, such as hunting, urbanization, agricultural and forestry expansion, and the introduction of exotic species, have intensified ecosystem disturbances, triggering a global biodiversity crisis (Dirzo et al., 2014; Oliveira et al., 2017). A less visible but equally concerning aspect of this crisis is the disruption of ecological interactions essential for ecosystem functioning, which may lead to functional extinctions and, eventually, to local or global species extinctions (Valiente-Banuet et al., 2015; Miguel, Cona & Campos, 2017).

In particular, the expansion of agriculture and urban infrastructure has simplified the structure of native forests, transforming them into ruderal grasslands and shrublands (Cramer, Hobbs & Standish, 2008). This structural degradation reduces native adult plant survival and habitat quality for seed germination and seedling recruitment, while favoring the establishment of exotic ruderal species that often compete with native flora (Traveset & Verdú, 2002; Bustamante, 2009; Escribano-Ávila et al., 2015; Torres, Castaño & Carranza-Quiceno, 2020). Additionally, habitat alteration increases predation risk and reduces food availability for specialized frugivores, particularly birds, thereby limiting plant reproductive success due to seed dispersal reduction (García, Zamora & Amico, 2011; Laborde & Thompson, 2013).

In human dominated-landscapes, mammals play a key role in seed dispersal. However, they are directly and negatively affected by habitat loss and fragmentation, which reduce their populations and, in turn, compromise seed dispersal processes (Napolitano et al., 2015; Cancio et al., 2016; Periago et al., 2017; Triay-Limonta et al., 2024). Although mammalian frugivory generally favors native plant persistence and habitat regeneration (Chama et al., 2013), it may also facilitate the spread of exotic invasive plant species, threatening the regeneration of native species, especially in landscapes dominated by invasive species (Richardson et al., 2000; Gosper & Smith, 2006; Traveset & Richardson, 2014; Leck, Parker & Simpson, 1989).

Seed dispersal effectiveness (SDE) consists of two main components: quantity, referring to visitation rates and number of seeds dispersed, and quality, referring to the viability, germination, and recruitment potential of those seeds (Schupp, Jordano & Gómez, 2010; Schupp, Jordano & Gómez, 2017). The effectiveness of the seed disperser, defined as its contribution to plant fitness, can be affected by anthropogenic pressures (Schupp, 1993; Schupp, Jordano & Gómez, 2010; Schupp, Jordano & Gómez, 2017). Habitat fragmentation restricts animal movement, reducing opportunities for plant-animal interactions and limiting seed dispersal distances and seedling recruitment (Galetti et al., 2006; Corrêa & Uriarte, 2012). Fewer visits by dispersers lead to reduced fruit removal and ultimately lower seed dispersal rates (Markl et al., 2012; Morán-López et al., 2015; Cancio et al., 2016; Periago et al., 2017). In addition, dispersal quality is influenced by factors such as seed viability and the impact of frugivore gut passage on germination (Traveset, Robertson & Rodríguez-Pérez, 2007; Torres, Castaño & Carranza-Quiceno, 2020). While for fleshy-fruited species gut passage can enhance germination through scarification, this effect varies across frugivores or plant species, and is also influenced by phylogenetic relationships between disperser and plant (Traveset & Verdú, 2002; Torres, Castaño & Carranza-Quiceno, 2020; Draper et al., 2022; Triay-Limonta et al., 2024). Anthropogenic disturbance can reduce the proportion of seeds that benefit from scarification and decrease chances of reaching suitable microsites for germination and recruitment (Traveset, Robertson & Rodríguez-Pérez, 2007; Escribano-Ávila et al., 2015).

Over the last 150 years, human activities have extensively altered the temperate forests of southern Chile (Smith-Ramírez & Armesto, 1994; Donoso, Ponce & Salas-Eljatib, 2018). Early colonization involved the widespread fire use for agriculture and cattle ranching, followed by intensive logging beginning in the 1920s. These actions led to the destruction of large areas of mature forests and the expansion of secondary forests (Otero, 2006; Donoso, Ponce & Salas-Eljatib, 2018), profoundly affecting forest composition, structure, and ecological function of forests at multiple levels (Otero, 2006). These forests support complex mutualistic networks between fleshy-fruited plants and frugivorous vertebrates, including birds, lizards, marsupials and foxes (Willson et al., 1996; Aizen, Vázquez & Smith-Ramírez, 2002; Smith-Ramírez & Armesto, 1994; Amico & Aizen, 2005; Amico & Aizen, 2000; Elgueta, Valenzuela & Rau, 2007; Jiménez, 2007; Zúñiga, Muñoz Pedreros & Fierro, 2008). Among mammals, three fox species (Lycalopex culpaeus, L. griseus and L. fulvipes) inhabit this region.

Current evidence suggests that the role of foxes as seed dispersers is complex and species-specific. They can enhance dispersal success in certain plants (e.g., Juniperus phoenicea; Farris et al., 2017), while in others they destroy most seeds during gut passage (e.g., Celtis australis; Rost, Pons & Bas, 2012). Their overall impact on germination remains unclear (Bustamante, Simonetti & Mella, 1992; Castro et al., 1994; León-Lobos & Kalin-Arroyo, 1994; Morales-Paredes, Valdivia & Sade, 2015). In Chile, frugivory has been documented for multiple plant species consumed by Lycalopex culpaeus, L. griseus, and L. fulvipes (Elgueta, Valenzuela & Rau, 2007; Jiménez, 2007; Torés, 2007; Zúñiga, Muñoz Pedreros & Fierro, 2008), with L. griseus reported as the most frugivorous species (Jaksic, Schlatter & Yáñez, 1980). However, despite this evidence, most seed dispersal studies have focused on birds (Markl et al., 2012), and it remains unclear how anthropogenic disturbances, such as habitat fragmentation and selective logging (Gray et al., 2007; Vargas et al., 2012), affect the role of South American foxes as dispersers of native and exotic plants in temperate forests.

In this study, we hypothesized that foxes effectively disperse exotic species in more disturbed landscapes and native species in less disturbed landscapes of southern Chile. We predict that: (1) feces containing exotic seeds will be more frequent in more disturbed landscapes, whereas feces with native seeds will be more common in less disturbed landscapes; (2) the number of exotic seeds will be higher in more disturbed landscapes, and the number of native seeds will be higher in less disturbed landscapes; (3) gut passage will not reduce seed viability; and (4) germination rates of exotic species will be higher in more disturbed landscapes, while native species seeds will germinate more successfully in less disturbed landscapes. The general objective of this study is to evaluate the seed dispersal effectiveness of foxes in temperate forests of southern Chile under varying degrees of landscape disturbance.

Materials & Methods

Study area and species

Fieldwork was conducted in the temperate forests of southern Chile, within and around national parks in the Los Lagos Region (Fig. 1), encompassing four different forest types defined by Luebert & Pliscoff (2023). All field activities within the National System of State-Protected Wild Areas (SNASPE) were conducted under the scientific research permit No. 005/2022 issued by the Corporación Nacional Forestal (CONAF), Región de Los Lagos, Chile.

Figure 1 Map showing the distribution of surveyed areas in southern Chile, including more disturbed and less disturbed landscapes.

National protected areas are shown in green. PN, National Park; RN, National Reserve.

The first forest stratum is the Andean temperate deciduous forest dominated by Nothofagus pumilio and Drimys andina, with frequent occurrences of fleshy-fruited shrubs such as Maytenus magellanica, Empetrum rubrum, Ribes cucullatum, Berberis serratodentada and B. montana, all of which bear fleshy fruits that attract frugivorous animals to disperse their seeds. The second stratum is the coastal temperate broadleaf forest, characterized by Weinmannia trichosperma and Laureliopsis philippiana, alongside Aextoxicon punctatum, Azara lanceolata, Eucryphia cordifolia, Rhaphithamnus spinosus and Raukaua laetevirens are also abundant. The third stratum is the inland temperate broadleaf forest, dominated by N. dombeyi and E. cordifolia, with additional species such as Persea lingue, Podocarpus salignus, W. trichosperma, L. philippiana and Dasyphyllum diacanthoides, among others. The fourth stratum is the Andean temperate evergreen forest characterized by N. dombeyi and Saxegothaea conspicua, as well as Podocarpus nubigenus, L. philippiana and W. trichosperma is also common (Luebert & Pliscoff, 2023). These forests support high diversity of trees and shrubs bearing fleshy fruits that attract frugivorous animals (Willson et al., 1996). The study area includes sites within the Puyehue, Alerce Andino and Vicente Pérez Rosales National Parks, as well as the Llanquihue National Reserve, representing less disturbed landscapes. It also includes privately owned parks, and areas designated for agricultural, forestry and livestock uses, which represent more disturbed landscapes. Three fox species inhabit the region: Lycalopex culpaeus, L. griseus and L. fulvipes, with the first two co-occurring at the study sites. All three are omnivorous and consume fleshy fruits, playing a role in seed dispersal across diverse ecosystems in Chile (Bustamante, Simonetti & Mella, 1992; Castro et al., 1994; León-Lobos & Kalin-Arroyo, 1994; Silva, Bozinovic & Jaksic, 2005; Elgueta, Valenzuela & Rau, 2007; Jiménez, 2007; Zúñiga, Muñoz Pedreros & Fierro, 2008; Morales-Paredes, Valdivia & Sade, 2015; Humaña & Valdivia, 2025). While L. culpaeus and L. griseus are widely distributed and use a range of habitats (Medel & Jaksic, 1988; Johnson & Franklin, 1994; Jaksic, 1997; Diuk-Wasser, 1995; Del Solar & Rau, 2004), L. fulvipes is an endemic species associated with native temperate rainforests (Jiménez, Lucherini & Novaro, 2008; Silva-Rodríguez et al., 2016; Silva-Rodríguez et al., 2018).

Habitat disturbance and sampling stations

To assess the effects of landscape disturbance on seed dispersal effectiveness (SDE), we first classified 10 sites qualitatively into two categories: less disturbed landscapes, characterized by large, continuous native forests and minimal human activity, and more disturbed landscapes, featuring fragmented forests and intensive land use. We then conducted a quantitative assessment by generating three km2 buffers around each fecal collection geographic point, in accordance with the home range described for the main species present in the study area, L. griseus (Silva-Rodríguez, Ortega-Solís & Jiménez, 2010). Within these buffers, we extracted 12 land-use variables from Mapbiomas Chile Collection (http://chile.mapbiomas.org/; Mapbiomas 2024), and obtained housing density data from the 2016 national census shapefile (INE, Instituto Nacional de Estadísticas, https://www.ine.gob.cl/). A total of 12 metrics were evaluated, including the percentage of agricultural mosaic, distance to the nearest dwelling, forest cover, number of forest fragments, among others (Table 1). These data were used to validate the disturbance classification of each site.

Table 1 Landscape variables that characterize and define more and less disturbed landscape types.

Variables	Less disturbed
Mean (SE)	More disturbed
Mean (SE)	
Agricultural and grazing mosaic (ha)	1.67 (0.3)	19.41 (5.6)	
Beach sand and dune (ha)	2.06 (0.5)	3.00 (0.6)	
Distance to nearest house (m)	1,091.07 (113.8)	867.79 (267,6)	
Exotic tree plantation (ha)	0.08 (0.02)	1.67 (0,6)	
Grassland (ha)	0.25 (0.06)	2.61 (4.36)	
Infrastructure (N∘ of Infrastructure / ha)	0.21 (0.08)	0.07 (0.1)	
Native forest (ha)	84.78 (1.9)	65.13 (4.8)	
Number of dwellings (N∘ of dwellings / ha)	9.21 (2.9)	20.26 (5.0)	
Number of forest patches	6.54 (0.8)	25.09 (1.9)	
Rocky outcrop (ha)	4.41 (1.07)	3.20 (0.5)	
Scrubland (ha)	<0.01 (0.001)	0.58 (0.1)	
Wetland (ha)	0.49 (0.1)	3.96 (0.7)	

From November 2022 to July 2023, we conducted systematic monthly surveys along 29 trails, each three to five kilometers in length and 0.7–1 meter wide, covering 86 kilometers per month and a total of 774 kilometers. Each trail was walked once a month for about two hours. Sampling effort was balanced across landscapes types: 19 trails were walked in less disturbed sites (seven in Puyehue, five in Alerce Andino, five in Vicente Pérez Rosales and two in Llanquihue) and 10 trails in more disturbed areas. Feces were collected with gloves and masks, split in half at collection site, one half was stored in paper bags at room temperature, the other preserved in 95% ethanol and frozen at −40 °C. GPS coordinates were recorded for each sample.

To characterize microsite conditions, we used one m2 quadrats centered on each fecal sample. We visually estimated canopy and ground cover following Soto & Puettmann (2018), and later used this data to analyze the influence of microsite conditions on seed germination.

Genetic identification of feces

DNA was extracted from collected feces using QIAamp DNA Stool Mini Kit (Qiagen, Hilden, Germany), following manufacturer protocols. Three mtDNA markers were amplified using PCR amplification (Saiki et al., 1985): (1) 16S rDNA (364 base pairs (bp)) (Hoelzel & Green, 1992; Johnson et al., 1998); (2) adenosine triphosphate ATP-8/ATP-6 genes (275 bp) with primers ATP8-DF1 and ATP6-DR1 and (3) the control region (CR) with the primers MTLPRO2 and CCR-DR1 (Tchaicka et al., 2007). PCR was carried out in 25 µL reactions containing 2.5 µL of buffer, 1.5 mM MgCl2, 0.2 mM of each dNTP, 0.75U of Platinum®Taq DNA Polymerase (Invitrogen, Waltham, MA, USA), 0.2 µM primers and 20–50 ng of DNA. Thermocycling was performed as described in Tchaicka et al. (2007). Products were visualized on 2% agarose gels stained with GelRed® and sequenced at Austral-omics (Universidad Austral de Chile), using an ABI 3500XL analyzer. Sequences were analyzed using Geneious Prime 2025.0.2.

Quantity of seed dispersal: seed identification and quantification

To evaluate the quantitative component of SDE, feces were dried at room temperature and seeds extracted via sieving and manual separation (Rubalcava-Castillo et al., 2020). Seeds were identified based on size, shape, and comparison with reference collections, with input from botanical experts and published sources (Smith-Ramírez & Armesto, 1994; Smith-Ramírez, Armesto & Figueroa, 1998). Seeds were classified as native or exotic. For each sample, we recorded the number of seeds per plant species.

Quality of seed dispersal: viability and germination

To estimate seed viability, we used a tetrazolium test on 10% of the seeds from each plant species present in a fecal sample. Seeds were sectioned and incubated in a 0.1% 2,3,5-triphenyltetrazolium chloride solution for 24 h at room temperature in the dark (Baskin & Baskin, 2004). Viable seeds displayed red staining of the embryo, indicating metabolic activity. Identification and counting were performed using a stereoscopic magnifying glass (Kyoto SMZ-140).

For germination trials, 10% of the seeds extracted were sown in the field at the original feces collection site. In total, 88 seeds of Aristotelia chilensis were sown (from 19 bags: 17 in less disturbed and two in more disturbed landscapes), 701 seeds of Gaultheria mucronata (27 bags: 26 in less disturbed and one in more disturbed landscapes), 1,069 seeds of Ugni molinae (55 bags: 48 in less disturbed and seven in more disturbed landscapes), five seeds of Malus sylvestris (two bags: one in less disturbed and one in more disturbed landscapes), 53 seeds of Prunus spp. (18 bags: nine in less disturbed and nine in more disturbed landscapes), and 390 seeds of Rubus ulmifolius (24 bags: 16 in less disturbed and eight in more disturbed landscapes). Seeds were placed in bags according to size: small (0.5–8 mm) in cloth bags (15  × 10 cm) and large (8–25 mm) in mesh bags (20  × 20 cm). Vegetation was cleared in a 50 cm radius; sites were marked with a metal stake and pink phosphorescent ribbon. Germination was monitored every 15 days from March 2023 to January 2024, by recording the presence or absence of a radicle of at least two mm in length considered a successful germination. Vegetation was cleared at each visit to standardize conditions.

Seed dispersal effectiveness

Seed dispersal effectiveness (SDE) was calculated for each plant species following Schupp, Jordano & Gómez (2010): SDE = Quantity × Quality. Quantity was defined as the number of feces with seeds multiplied by the number of seeds per plant species. Quality was calculated as the product of seed viability and germination, adjusted for microsite conditions (canopy and ground cover). This approach allowed us to assess the influences of both landscape-level disturbances and small-scale environmental variations on dispersal performance.

Statistical analysis

The assumptions of several linear models with Poisson error distributions for seed number, viability and SDE, and Gaussian error distributions for germination were previously tested. These were evaluated and, in the case of the former, none of them met the assumptions. In the case of germination percentage, several linear models met the assumptions and the one with the lowest AIC value was selected. ANOVA followed by Tukey’s HSD was used to test differences between native and exotic plant species, landscape disturbance, and microsite covariates (canopy cover). To evaluate the effects of landscape disturbance (more and less disturbed) and plant species (A. chilensis, G. mucronata, U. molinae, M. sylvestris, R. ulmifolius, Prunus spp.) on number of seeds and viability, we performed a PERMANOVA analysis with 999 permutations. Pairwise contrasts were used to test differences between plant species and by landscape type. Finally, Mann–Whitney tests were used to compare SDE values between plant species within less and more disturbed landscapes separately. All statistical analyses were performed in R version 4.1 (R Core Team, 2024).

Results

Habitat disturbance and sampling stations

In terms of land use, agricultural and grazing mosaic and exotic tree plantations were 11.6 and 20.9 times higher in the more disturbed than in the less disturbed landscapes (Table 1). Regarding human occupation, the distance to nearest house and infrastructure were 1.3 and 3 times higher in less disturbed than more disturbed landscapes, while the number of dwellings was 2.2 times higher in more disturbed than less disturbed landscapes (Table 1). Finally, in terms of landscape physiognomy, the cover of beach sand and dunes, grassland, number of forest patches, scrublands and wetlands were 1.5, 10.4, 3.8, 58 and 8.1 times higher, respectively in more disturbed than less disturbed landscapes. In contrast, native forest and rocky outcrops were 1.3 and 1.4 times higher in less disturbed than in more disturbed landscapes (Table 1).

Sampling and genetic identification of feces

Between November 2022 and July 2023, a total of 199 feces were collected from all study sites. Of these, 131 samples (65.8%) contained seeds, with 130 of the 131 identified at least to genus level (99.2%); 97 corresponded to Lycalopex griseus (74.6%), while 4.7% were attributed to L. culpaeus and 18.9% could only be identified to genus level Lycalopex spp. Three fecal samples of non-target species were excluded from further analysis (Table 2). Due to limited species-level identification in some cases, all fox fecal samples were subsequently analyzed collectively as Lycalopex spp.

Quantity of seed dispersal: Seed identification and quantification

Sampling effort was consistent across all landscapes, with monthly coverage in both more disturbed and less disturbed areas. Seed-bearing fecal samples were detected only between December 2022 and June 2023, with the highest frequency recorded in February 2023. No seeds were found in fecal samples collected in November 2022 or July 2023 (Fig. 2).

A total of 23,012 seeds from six plant species were recovered from fecal samples. Native species included the tree Aristotelia chilensis (seed size: 2–3 mm; OT Limonta, pers. obs., 2024) and the shrubs Gaultheria mucronata (0.5–0.9 mm; OT Limonta, pers. obs., 2024) and Ugni molinae (1.5–2 mm; OT Limonta, pers. obs., 2024). Exotic species comprised the trees Prunus spp. (8–25 mm; Muñoz et al., 2001) and Malus sylvestris (6–8 mm; OT Limonta, pers. obs., 2024), and the shrub Rubus ulmifolius (1.5–3 mm; OT Limonta, pers. obs., 2024) (Fig. 3). Seeds that could not be taxonomically identified to the species level were excluded from statistical analyses.

Table 2 Genetic identification of the species of origin of the feces sampled at the study sites according to landscape type.

Fox species	More disturbed landscapes	Less disturbed landscapes	Total	
Lycalopex culpaeus	0	6 (4.7%)	6 (4.7%)	
Lycalopex griseus	19 (14.9%)	78 (61.4%)	97 (76.4%)	
Lycalopex spp.	3 (2.4%)	21 (16.5%)	24 (18.9%)	
Total	22 (17.3%)	105 (82.7%)	127 (100%)	

Figure 2 Quantity of seed-containing and seed-free fecal samples found at the study sites during the sampling period (November 2022 to July 2023).

Figure 3 The chilla fox is the most common fox in the study area, both in the more and less disturbed areas, and feeds on the fruits of native and exotic wild plants, as well as cultivated plants not shown here, such as apples (Malus sylvestris) and cherries.

(A) Lycalopex griseus, (B) the native tree Aristotelia chilensis, (C) the native shrub Gaultheria mucronata, (D) the exotic and invasive shrub Rubus ulmifolius, and (E) the native shrub Ugni molinae. Photo credits: Carlos E. Valdivia.

The number of fecal samples containing seeds of native plants was notably higher in less disturbed landscapes. Specifically, samples containing seeds of A. chilensis, G. mucronata, and U. molinae were 8, 21, and 5 times more frequent, respectively, in the less disturbed areas compared to more disturbed areas (Fig. 4A). In contrast, patterns for exotic species were less consistent: feces containing seeds of R. ulmifolius were 1.5 times more frequent in less disturbed areas, while the occurrence of seeds from M. sylvestris and Prunus spp. was similar across both landscape types (Fig. 4B).

Figure 4 (A) Distribution of the number of fox fecal samples containing seeds of native and (B) exotic plant species in more and less disturbed landscapes. (C) Number of seeds per fecal sample of native and (D) exotic plant species in the same landscapes.

Boxes represent the interquartile range (IQR), the central line indicates the median, and whiskers extend up to 1.5 times the IQR from the quartiles. Individual points represent outliers. Different letters indicate statistically significant differences (p < 0.05).

When analyzing the number of seeds per fecal sample, a two-way PERMANOVA revealed no significant effect of landscape type (F = 1.8, p = 0.18) or landscape type × species interaction (F = 0.9, p = 0.46). However, a significant effect of plant species was observed (F = 5.5, p < 0.01). Among native plants, U. molinae had 1.4 times more seeds per fecal sample in more disturbed areas, whereas G. mucronata showed 4.3 times more seeds per fecal sample in less disturbed areas. For A. chilensis, no significant differences were observed between landscape types (Fig. 4C). Regarding exotic species, R. ulmifolius seeds were 1.7 times more numerous in the less disturbed areas. No significant differences were found for M. sylvestris and Prunus spp. between landscape types (Fig. 4D).

Quality of seed dispersal: Seed viability and germination

Seed viability varied significantly among species (PERMANOVA: F = 10.1, p < 0.05), but not according to landscape type (F = 0.005, p = 0.94) or the interaction between landscape × species (F = 0.9, p = 0.5). The viability of A. chilensis and G. mucronata seeds was 1.2 and 1.1 times higher, respectively, in less disturbed landscapes, whereas U. molinae showed slightly higher viability (1.1- fold) in more disturbed areas (Fig. 5A). Among exotic species, seed viability of M. sylvestris and Prunus spp. showed no significant differences between landscapes, while R. ulmifolius was 1.1-fold higher in more disturbed landscape (Fig. 5B).

Figure 5 (A) Seed viability percentage of native and (B) exotic plant species, and (C) seed germination rates of native and (D) exotic plant species in more and less disturbed landscapes.

In the boxplots, the central line within each box represents the median, and the upper and lower box edges indicate the third (Q3) and first (Q1) quartiles, respectively, encompassing the central 50% of the data. Whiskers extend to the last data point within 1.5 times the interquartile range (IQR), and individual points represent outliers. Different letters indicate statistically significant differences (P < 0.05).

Seed germination rates varied by species and landscape type. Due to an extreme weather event that washed away the germination bags of M. sylvestris, no data were available for this species. Germination of G. mucronata failed to germinate in more disturbed landscapes, despite exhibiting high initial viability (80%). ANOVA revealed significant effects of landscape type (F = 26.5, p < 0.001), plant species (F = 17.6, p < 0.001), canopy cover (F = 6.15, p < 0.001) and the species x landscape interaction (F = 10.5, p < 0.001). Among native species, germination rates of A. chilensis and U. molinae did not differ significantly between landscapes type, whereas G. mucronata germinated only in the less disturbed areas (Fig. 5C). Regarding exotic species, Prunus spp. germinated at similar rates in both landscapes types, while R. ulmifolius showed 1.6 times higher germination in the more disturbed landscapes (Fig. 5D).

Seed dispersal effectiveness

Seed dispersal effectiveness (SDE), was calculated as the product of quantity and quality components, and it varied among species and landscapes types. Mann–Whitney U tests revealed no significant differences in SDE for Prunus spp. (U = 9, p = 0.882, r = 0.05), and R. ulmifolius (U = 15, p = 0.149, r = 0.48). In contrast, U. molinae showed significantly higher SDE in more disturbed landscapes (U = 41, p = 0.016), with a large effect size (r = 0.64). These results suggest that while some species maintain consistent dispersal effectiveness across landscape types, others—such as U. molinae—may experience substantial shifts in dispersal dynamics depending on the degree of anthropogenic disturbance (Table 3). Due to an extreme weather event that destroyed the germination bags of A. chilensis in the more disturbed landscapes, data were available from only a single sample; hence, no statistical analysis was conducted for this species. Similarly, G. mucronata failed to germinate under the more disturbed landscape conditions, precluding statistical comparison.

Table 3 Seed dispersal effectiveness (SDE) native and exotic plant species dispersed by foxes in more disturbed and less disturbed landscapes.

Plant species	SDE	
	More disturbed landscapes
Mean (SE)	Less disturbed landscapes
Mean (SE)	
(a) Native			
Aristotelia chilensis	6,060.61 (0)	468.1 (84.7)	
Gaultheria mucronata	0 (0)	306.5 (238.5)	
Ugni molinae	1,914.2 (597.4)	285.6 (68.6)	
(b) Exotic			
Prunus spp.	725.1 (488.9)	714.3 (566.8)	
Rubus ulmifolius	113.4 (58.8)	11.24 (11.3)	

Discussion

Our results showed that seeds appeared in fox feces mainly between December and June, peaking in February. This seasonal pattern aligns with previous studies reporting increased fruit consumption by foxes during spring and summer, when fruit availability is highest (Castro et al., 1994; Zúñiga, Muñoz Pedreros & Fierro, 2008; Zúñiga, Fuenzalida & Sandoval, 2018; Zúñiga et al., 2020; Zúñiga et al., 2022). The absence of seeds in November and July likely reflects low fruit availability during these months, suggesting a strong link between plant phenology and fox frugivory (Dalponte & De Souza, 1999; Aragona & Setz, 2001; Varela et al., 2008). However, seasonal consumption patterns may vary geographically, depending on prey availability and habitat type (Silva, Bozinovic & Jaksic, 2005; Rubalcava-Castillo et al., 2020). The dietary flexibility allows foxes to adapt to temporal variations in resource availability, and in doing so, directly influences seed dispersal dynamics and plant regeneration throughout the year. Future studies that consider the full annual cycle of fruit and prey availability will help to better understand these dynamics and their consequences for forest regeneration (Kidawa & Kowalczyk, 2011; Villalobos, Buenrostro-Silva & Sánchez-de la Vega, 2014).

A total of 23,012 seeds belonging to six species were recovered, including native (Aristotelia chilensis, Gaultheria mucronata, Ugni molinae) and exotic (Rubus ulmifolius, Malus sylvestris, Prunus spp.) species. Most seeds remained intact after gut passage, likely due to their small size and resistant pericarp (Traveset & Verdú, 2002; Torres, Castaño & Carranza-Quiceno, 2020; Draper et al., 2022). This confirms that foxes can act as effective dispersers of plant species with small to medium-sized fleshy fruits that rely on endozoochory for their reproduction (Peredo et al., 2013; Amodeo, Vázquez & Zalba, 2017). From an ecological perspective, seed dispersal by carnivores facilitates the colonization and persistence of plant species across different habitats, enhancing connectivity among plant populations and contributing to both the structural and functional diversity of the forest.

The comparison between landscapes with different degrees of human disturbance showed clear differences in seed dispersal patterns. Seeds from native plant species were significantly more prevalent in feces collected from less disturbed sites, up to 21 times more for G. mucronata. This pattern likely reflects a greater availability of native fruits in less altered landscapes, along with potentially higher dispersal efficiency by foxes in these environments (Armesto et al., 1987; Sabag, 1993). In contrast, the dispersal of exotic species such as R. ulmifolius was relatively high across both landscape types, suggesting that these invasive plant species are well established and widely available for consumption even in less disturbed areas. This may facilitate their continued spread, with negative consequences for native flora (Cifuentes, 2018; Rejmánek, 2015). Ecologically, these patterns have important implications for plant community structure and dynamics. The reduced dispersal of native seeds in disturbed landscapes could limit natural regeneration and forest resilience, potentially favoring exotic species and contributing to biodiversity loss (Williams & Ward, 2006; Tavşanoğlu et al., 2021). Seed dispersal by foxes may thus act as a mechanism for either community maintenance or transformation, depending on the landscape context and species involved. This phenomenon highlights the ecological complexity of plant-disperser interactions in fragmented landscapes, where anthropogenic disturbances reshape trophic networks and regeneration and alter regeneration processes.

Analysis of the number of seeds per fecal sample revealed species and landscape specific patterns, indicating that foxes modulate their diet according to fruit availability and the degree of disturbance. For example, U. molinae seeds were more abundant in feces from more disturbed areas, while G. mucronata predominated in less disturbed areas. This pattern may reflect dietary flexibility, allowing foxes to adapt to fluctuations in food resources, contributing to seed dispersal under varying environmental conditions and potentially facilitating the persistence of certain native species under moderate disturbance (Amico, Rodríguez-Cabal & Aizen, 2009; León-Lobos & Kalin-Arroyo, 1994). The inclusion of foxes as effective dispersers of U. molinae broadens our knowledge of dispersal networks in these ecosystems. Traditionally attributed to marsupials such as Dromiciops gliroides, seed dispersal by carnivorous mammals suggests greater ecological complexity and functional redundancy, which could confer greater stability to the dispersal process (Amico, Rodríguez-Cabal & Aizen, 2009). This redundancy is important for biodiversity conservation, especially in the context of loss or decline of specific dispersers (Stevenson, 2011). However, the dispersal of exotic seeds such as R. ulmifolius by foxes poses a conservation challenge, as it highlights their dual role in both promoting native forest regeneration and facilitating invasive species spread (Dellafiore, Brignone & Scilingo, 2020). In this context, managing invasive plant species requieres considering the interactions between flora and dispersing fauna in both protected areas and disturbed landscapes.

The viability of seeds after passing through the digestive tract is essential for germination, survival, and seedling establishment (Rubalcava-Castillo et al., 2023). Our results confirm this premise: seeds dispersed by foxes of the native species A. chilensis, G. mucronata and U. molinae, as well as the exotic species M. sylvestris, R. ulmifolius and Prunus spp., remained viable after ingestion. Five of these species exhibited higher seed viability in less disturbed landscapes, with the exception of M. sylvestris, which showed 100% viability in both types of landscapes. These findings contrast with previous studies suggesting that foxes of the genus Lycalopex could be illegitimate dispersers, reducing seed viability or germination (Bustamante, Simonetti & Mella, 1992; Castro et al., 1994; León-Lobos & Kalin-Arroyo, 1994; Morales-Paredes, Valdivia & Sade, 2015; Dellafiore, 2018; Maldonado et al., 2018; Vergara-Tabares, Whitworth-Hulse & Funes, 2018; Duarte & Dellafiore, 2020; Duarte & Dellafiore, 2021). Most seeds that pass through carnivore digestive tracts are found intact in feces, and the effect on germination and dormancy is variable and species-dependent (Draper et al., 2022). In some cases, gut passage may break dormancy or modify germination, either accelerating or delaying it (Traveset & Wilson, 1997; Rubalcava-Castillo et al., 2021).

In our study, the seeds of most native and exotic plants germinated adequately in both types of landscapes, except for G. mucronata, which did not germinate in the more disturbed landscapes despite an initial viability of 80%. This suggests that the characteristics of microsites in disturbed environments may not be favorable for germination (Traveset & Verdú, 2002; Torres, Castaño & Carranza-Quiceno, 2020). This result was unexpected, considering that species of the genus Gaultheria, particularly G. mucronata and G. phyllireifolia, are often found in partially disturbed scrublands following logging in temperate forests (Luebert & Pliscoff, 2023). One possible explanation is that Gaultheria forms seed banks in the soil, with seeds that can remain dormant for several years before germinating (seeds with prolonged dormancy). The Chilean endemic shrub U. molinae has two propagation systems, sexual and vegetative, which provides it with greater flexibility for establishment and perpetuation. Seeds of this Myrtaceae species are dispersed via endozoochory and typically germinate in natural environments after approximately five weeks, although they have lower germination compared to other Myrtaceae (Figueroa, 2003). Previous studies suggest that high viability combined with low germination indicates strong dormancy (Smith-Ramírez, Armesto & Figueroa, 1998), although in our study we observed high germination, probably influenced by the favorable conditions of the microsites where they were deposited. A. chilensis, commonly referred to as maqui, is a tree that is native to the temperate forests of Chile and Argentina, also reproduces sexually and vegetatively, and its germination is significantly higher in forest clearings compared to under closed canopy (Hoffmann et al., 1992; Moesbach, 1992; Figueroa, 2003). In addition, forest fragmentation negatively affects seed germination, as seeds of plants from smaller fragments show lower germination rates (Valdivia & Simonetti, 2007). Our results are consistent with this trend, showing higher viability and germination in less disturbed landscapes. As for R. ulmifolius, its fruits are consumed abundantly by birds and mammals, which disperse its seeds and they remain viable in the soil, with germination linked to environmental disturbances (Cifuentes, 2018). In the Araucanía Region, R. praecox has been observed to establish itself under the canopy of A. chilensis, facilitated by the partial coincidence of their fruiting periods and dispersal by birds and foxes (Rejmánek, 2015). The invasion of R. ulmifolius hinders the regeneration and growth of native species, affecting the composition, structure and natural dynamics of forests and scrublands, as well as contributing to forest fragmentation (Cifuentes, 2018). The seeds of most Prunus species are dormant when they mature and require prolonged cold or warm stratification to break dormancy, due to the restriction of the endocarp on embryo emergence (Chen et al., 2007; Imani et al., 2011). For example, P. campanulata seeds stratified in cold and humidity germinate at high rates (Chen et al., 2007). This technique has been widely used to maximise the germination of seeds with water-permeable endocarp, such as those of Prunus (Schopmeyer, 1974; Nikolaeva, 1977; Baskin & Baskin, 2004). The germination test in our study encompassed winter, which probably favored stratification due to low temperatures and high humidity, increasing the probability of germination.

Our findings support the role of Lycalopex foxes as effective seed dispersers of both native and exotic plant species in southern Chilean forests. Although seed dispersal effectiveness (SDE)—the product of quantity and quality—did not show consistent differences between landscape types, notable patterns emerged that reflect species-specific responses to disturbance. For A. chilensis, although data limitations in more disturbed areas prevented statistical comparison, observed values suggest higher seed dispersal in more disturbed landscapes. As this species is less consumed by frugivorous birds in disturbed habitats (Valdivia & Simonetti, 2007), foxes may compensate for this functional gap, enhancing seed input. Given its high phenotypic plasticity and preference for high-light conditions, A. chilensis may benefit from dispersal in open, disturbed sites. In the case of G. mucronata, no germination occurred in the more disturbed landscapes, but seeds germinated successfully in less disturbed sites, indicating partial dispersal effectiveness by foxes in favorable conditions. Its prevalence along roadsides and forest edges (Luebert & Pliscoff, 2023), which are frequently used by foxes, likely facilitates seed consumption and deposition. This spatial overlap may promote stable dispersal interactions in semi-natural landscapes, although harsher conditions in more disturbed sites may limit germination success regardless of seed arrival. In contrast, U. molinae, by contrast, exhibited a clearly higher SDE in more disturbed areas, likely due to a combination of greater seed deposition by foxes and favorable germination conditions. This species often grows in forest edges and shrublands frequented by foxes (Figueroa, 2003; Smith-Ramírez, Armesto & Figueroa, 1998), enhancing spatial overlap between plant and dispersers and seed deposition in suitable microsites. Additionally, its seeds may be more tolerant to the abiotic stress of altered habitats, further contributing to effective recruitment (Schupp, Jordano & Gómez, 2010; Schupp, Jordano & Gómez, 2017).

The exotic and invasive R. ulmifolius showed a pattern of higher SDE in more disturbed areas, likely due to its ecological strategy: a preference for open, high-light environments and regeneration via both seeds and vegetative parts. These traits, couple with fox-mediated dispersal, could support its spread across fragmented or anthropized landscapes (Rejmánek, 2015; Cifuentes, 2018). Such interactions between generalist dispersers and invasive species can have long-term consequences for native plant communities and ecosystem dynamics. Finally, Prunus spp. displayed comparable SDE across both landscape types, though germination was slightly higher in less disturbed areas. This difference may be explained by local environmental conditions—such as increased light availability and reduced competition—that favor seedling establishment in mature forest patches (Savill, 1971; Kerr & Evans, 1993). Despite not being invasive in Chile (Arriagada, 1987; Loewe, 1991; Vergara, 1991), its wide cultivation and adaptability raise the possibility of naturalization facilitated by animal dispersers.

Taken together, these findings reinforce the insight that the effectiveness of seed dispersal by Lycalopex foxes varies by plant species and landscape contexts. Despite being dietary generalists, foxes show effective dispersal patterns that may be modulated by fruit availability, habitat use, and environmental characteristics. As potential functional compensators in anthropized landscapes—where other dispersers may be absent or less effective—foxes may play a key role in both native plant regeneration and exotic spread. These findings broaden our understanding of seed dispersal by mammals, a topic still underexplored beyond primate’s studies (Triay-Limonta et al., 2024), and emphasize the importance of jointly considering plant ecological traits and landscape disturbance when evaluating dispersal dynamics.

Supplemental Information

Supplemental Information 1 Raw data

The raw data include seeds extracted from fox feces collected across different landscapes, with information on seed identification and quantity per plant species, classified as native or exotic. The dataset also contains variables derived from satellite imagery describing the landscape (number of forest patches, distance to dwellings, land cover types, infrastructure, among others) and the classification of landscapes as less or more disturbed. Additionally, experimental data on seed viability and germination are included, which were used to calculate the Seed Dispersal Effectiveness (SDE) index for each plant species and landscape type. These data were used for the statistical analyses presented in this study.

The authors of this article would like to thank the various institutions that supported us in conducting this research. We are grateful to the National Forestry Corporation (CONAF) of the Los Lagos region in Chile, to the private parks Valle Los Ulmos (Ensenada, Chile) and Katalapi. O. T.-L. is grateful for the collaboration received from Dr. Jaime Rau and Dr. Edwin Niklitschek. The author O.T.-L- would like to acknowledge the use of Grammarly AI, which she used for the editing process, identifying problems of style, grammar and structure in the manuscript, speeding up the editing process and increasing the final quality of the product, thanks to this advanced language processing tool’s ability to detect errors and suggest improvements.

Additional Information and Declarations

Competing Interests

Author Contributions

Field Study Permissions

Data Availability

The authors declare there are no competing interests.

Onaylis Triay-Limonta conceived and designed the experiments, performed the experiments, analyzed the data, prepared figures and/or tables, authored or reviewed drafts of the article, and approved the final draft.

Rocío Paleo-López conceived and designed the experiments, performed the experiments, analyzed the data, authored or reviewed drafts of the article, and approved the final draft.

Camila J Stuardo conceived and designed the experiments, performed the experiments, analyzed the data, prepared figures and/or tables, authored or reviewed drafts of the article, and approved the final draft.

Carolina S Ugarte conceived and designed the experiments, performed the experiments, analyzed the data, authored or reviewed drafts of the article, and approved the final draft.

Carlos E. Valdivia conceived and designed the experiments, performed the experiments, analyzed the data, prepared figures and/or tables, authored or reviewed drafts of the article, and approved the final draft.

Constanza Napolitano conceived and designed the experiments, performed the experiments, analyzed the data, prepared figures and/or tables, authored or reviewed drafts of the article, and approved the final draft.

The following information was supplied relating to field study approvals (i.e., approving body and any reference numbers):

The field experiments were approved by the National Forestry Corporation of Chile.

The following information was supplied regarding data availability:

The raw data are available in the Supplementary File.

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
