# Peer review of "Effectiveness of seed dispersal by foxes in areas with different human disturbances in southern Chile"

_PeerJ, doi:10.7717/peerj.20150_

## Round 0.1 · original submission · Major Revisions

Thank you very much for your manuscript titled “Effectiveness of seed dispersal by foxes in areas with different human disturbances in southern Chile” that you sent to PeerJ.

This study presents very valuable and relevant information on the phenomenon of seed dispersal by mammals as a function of environmental disturbance.
As you will see below, comments from referee 1 suggest a minor revision while reviewers 2 suggests a major revision before your paper can be published. Given this, I would like to see a major revision dealing with the comments. Their comments should provide a clear idea for you to review, hopefully improving the clarity and rigor of the presentation of your work. I will be happy to accept your article pending further revisions, detailed by the referees.

Reviewer 1 suggests reducing the introduction, including citations, as well as the methods and discussion, and improving the editing of the figures.

Reviewer 2 suggests that the context of the abstract and introduction be improved, with a clearer statement and objectives and improved grammar of the text. It is also suggested that several points of the experimental design be clarified and that the discussion be focused on the relevance of the results obtained.

Please note that we consider these revisions to be important and your revised manuscript will likely need to be revised again.

Reviewer 1 ·

Basic reporting

In a nutshell, the authors evaluated variation in seed dispersal effectiveness (SDE) by Lycalopex foxes for native and exotic plant species in disturbed and undisturbed areas of southern Chile. They analyzed SDE considering both quantity and quality. The former included the frequency of feces with seeds and the number of seeds per plant species consumed, while the latter assessed the proportion of viable seeds after throughput and their germination capacity. Their results showed that foxes dispersed seeds of three native species (Aristotelia chilensis, Gaultheria mucronate, Ugni molinae) and three exotic ones (Malus sylvestris, Rubus ulmifolius, Prunus spp.). SDE was higher for the former two in undisturbed landscapes, while it was higher for the latter in disturbed areas. Foxes were more effective in dispersing seeds of Prunus spp. in undisturbed landscapes and of R. ulmifolius in disturbed areas.

This paper is well written and thus easy to follow, but excessive details tend to obscure the most important messages. There is excessive citation, sometimes rather dated, to refer to what is already accepted knowledge of the phenomenon addressed (frugivory and dispersal by vertebrates). For instance, the first five paragraphs of the Introduction, should perfectly fit in a single one. The relevant contribution is within the two following paragraph. As it is, the Intro looks like the opening chapter of a thesis.

The Methods are painstakingly detailed, which I appreciate, but could be shortened if needed. The Results are also very detailed, and I have only one observation: In line 378, I wonder if an F = 0.1 can yield p < 0.05. The Discussion is interesting, but also rather long. I wonder if it could be synthesized, not citing every other study conducted in different continents and with different carnivore species (Bears in Europe? Coyotes in N. America?). Because the Discussion is already so long and disperse, it warrants a Conclusion section. But if the Discussion were tightened, it would not. The References, in excess of 200 should be thinned out to the most relevant ones.

Among figures, I note that Fig. 1 duplicates the name “Los Muermos.” In Fig. 4, I wonder what is the meaning of the dots in panels C and D; if the bars depict 1 SD, then what is the meaning of the whiskers? Ditto Fig. 5. l have no problem with the three Tables.

Experimental design

No problem

Validity of the findings

I concur. Check an F value.

Additional comments

Shorten it

Reviewer 2 ·

Basic reporting

The main problems I found with the manuscript are in the writing and presentation (i.e., the basic reporting). All problems are solvable, although I feel the changes required are major. I am listing the relevant major comments here and minor comments under "Additional comments".

The introduction needs considerable editing to improve structure, reduce repetition and references and to ensure there is sufficient context to understand the goals of the study. (1) The authors provide predictions on where they expect exotic vs. native species to be dispersed to, but there is no background in the introduction to explain their predictions. For example, do the foxes inhabit both habitat types and possibly move between them? What do we actually know about their diet in relation to exotic vs. native fruits? (2) the structure of the paragraphs need improvement. All seem to introduce themes of disturbance and mutualism/seed dispersal so it is hard to figure out the main message of each paragraph. Care needs to be taken to remove repetition and ensure a clear point is made. (3) there is also heavy use of references – probably far more than needed. I can understand the need to reference previous fox papers in full, but other points only need sufficient references to support the point. Note that the discussion has the same problem

The grammar and phrasing needs some work. There are grammatical errors and also odd phrasing (“imperative to comprehend”, “grows the trees”). I expect the phrasing is AI suggested, but perhaps it is possible to ensure suggestions are scientific, or more simple?

The authors often rely on statements of data gaps to defend their study. PeerJ does not have a novelty requirement so a data gap is not essential to have. Also, it is not useful to simply state there is a gap as not all gaps need to be studied! It is better to explain clearly why it is important to know something and – with a clear study context – the “gap” should become apparent.

Do you have information on seed sizes? This is important for understanding the quantity of seeds in the feces. Your comparisons are within each species so there is no scope for error, but if you can provide seed size information it would be good.

Experimental design

No clear aims or goals of the study are given. There are predictions which are very useful, but there also needs to be well defined aims.

The authors defined disturbed and undisturbed habitats, and then measured a series of disturbance attributes surrounding the scats. No where have they explained how they integrated these two measures and what the measurements were actually used for. In the results, they compare the measured attributes between what they defined as disturbed and undisturbed but the reason for these results is not clear. This part of the results also has no clear associated aim and seems more methodological. Were the attributes measured for better definition of what a disturbed habitat is in relation to the scats? There needs to be better clarity on why these were done and how they are connected to the study’s objectives.

There are other problems in the methods such as not reporting clearly whether search area is equivalent across habitats (disturbed/undisturbed) and months. This directly impacts the validity of the results for scats so should be made clear. If sampling intensity is variable then the results need to be adjusted for area.

Validity of the findings

The discussion contains lots of interesting information, but some of it is more “review-like” and fails to make a clear discussion point. The point of the discussion is to explain the results, but there are long lists of dispersal agents (for example) for the plant species found in the fox scats. Some thought needs to go into the key points that are needed to explain and understand the results and avoid a mini-review. It is also important to discuss the main objectives early in the discussion: this was the patterns of exotics vs natives in disturbed and undisturbed areas. The first paragraph overviewing the results is good. The second is a good paragraph but should perhaps come after a paragraph that comes back to the goals/predictions of the study. Overall, the discussion needs work on structure.

In the results, I understand that of the 131 feces with seeds, 130 were identified to species level. Why then, are the results not presented at species level? Only a single feces could not be identified which is fine. But, then Table 2 shows something different with 24 not identified to species. Something is not correct in the writing.

Additional comments

The abstract fails to define the major context of the study which is fox dispersal of native and exotic species across disturbed and undisturbed landscapes.

L55: I think you mean secondary extinctions here, but functional extinction has broader impacts than just this so it is quite a jump, and not well explained.

L65-67: Again, this is a single scenario but is written as though it occurs in all cases.

L76-77: What do you mean by mostly mutualistic mammals? That they are the main dispersers? What about birds?

L78-82: This topic does not seem to fit here. But, generally, I feel the introduction needs work in structure.

L103-105: This seems to be a bit off-topic? (not directly related to what I think the aims are).

L116-122: The reasoning here is odd as quality is related to the specific interaction between animal and plant and independent of quantity. Are you saying that animal-treatment of seeds will differ between habitats?

L123: This paragraph. Be sure to connect this to the aims of the study. Some of it reads more like a review and is perhaps more appropriate in the study site info.

L133: But this is true of all forests isn’t? The wording is odd and suggests the forest is special to have this.

L166: “strata” - I think you mean forest type? (and also rest of paragraph).

L233: Is this the length of each trail or a total? Can you confirm in the text that the same length of transect was walked each month? This is important for some of the results as they are not corrected for search area. Did you search the same area for both disturbed and undisturbed habitats? This information is also needed and if not equal a correction will be needed.
Also, I feel this paragraph should come before you explain landscape types.

L250: State clearly that this was done to identify disperser.

L336: Are the foxes similar in body size and habitat use? It would be good to provide this information so we can understand how valid it is to combine species.

L349: Can you give a number rather than “few”?

L383: But this paragraph is about viability and not germination.

L511: the topic sentence here is odd and a clear point is not presented in the paragraph.

L578: I was very confused by this point at first. Make it clear you are talking about quantitative measures of SDE. However, does it matter that there is only one prior study on foxes? Does it matter that there are more studies on primates (which are major frugivores in tropical forests and are justifiably better studied)? Be careful that this paragraph addresses a clear and important point and not just highlights novelty that may or may not be important. Some of the points raised later are important but the topic sentence does not clearly describe what the paragraph is about, and the paragraph wanders too much across topics.

L620: “dispersal failure” is a bit of a conceptual jump from the current study! There is no need to overstate the value and application of your work. Your study is interesting and important to publish.

L623: Not all frugivores are generalists with long distance movements….

---

## Round 0.2 · Minor Revisions

After reviewing this revised version of your manuscript, I see that the main comments suggested by the reviewers have been included. However, there are still some details that need to be clarified before having a final version that can be published.

It is necessary to check the English and the discussion, in addition to some editing details.

Reviewer 2 ·

Basic reporting

The manuscript has improved considerably since the first version and I only found minor problems with this version. Some English errors remain, so it is worth checking it again with a language-editing tool.

The discussion begins with a "review" format rather than highlighting the main findings. It is preferable to have the latter as this is a key place the reader looks to understand the study "in a nutshell". However, it is the Editors choice as to whether it needs to be corrected or not. It also reads fine as it is.

Corrections:
L25: What do you mean by "plant origin"? The phrasing is not clear.

L28: "more and less disturbed" I don't like this phrasing, but I understand why the authors use it. Perhaps the Editor can decide if it should be changed or not. It would be good to mention the number of sites you studied in.

L104-105: Are all these refs for foxes? It is quite a lot - so, what do you consider it still poorly understood? Why do you expect them to be important? These two points need a brief mention.

L138-139: This needs a reference

L143: Would be good to have the phrasing to indicate that the occur together (or not) in the same areas.

L157: Give the number of sites here.

L160: Why 3 km? Can you give a reason?

L164: State how many metrics you included here.

L167-168: How many km did you walk each month (in total) and how many in total overall? Please give this info.

L176: Do you need the word "quality" here? Because what makes a high quality site will differ among species (as you found).

L178: State clearly it was from feces (i.e., text should be clear without looking at the subheading).

L204: What is the maximum size for a "small" seed?

L313: My apologies if I missed this, but somewhere, you need to indicate how many independent germination tests there were per species. That is how many feces were germinated per species.

L470: It would be good to specify what environment you are referring to here. It is not immediately obvious.

Experimental design

The experimental design remains the same as before and is fine.

Research question is well defined and meaningful. In my comments above I request a little more detail on the knowledge gap.

Methods are described well with some minor issues described in my comments above.

Validity of the findings

I have not checked the data.

Conclusions are well stated and in line with the results.

---

## Round 0.3 · accepted · Accept

After reviewing this revised version of your manuscript, I see that the comments suggested by the reviewers have been included. Therefore, I am satisfied with the current version and consider it ready for publication.